# "Everything was much more dynamic": Temporality of health system responses to Covid-19 in Colombia

**Simon Turner** ⓘ *, **Dennys Paola Fernandez**

School of Management, University of los Andes, Bogotá, Colombia

* s.turner@uniandes.edu.co

**Data Availability Statement:** This paper draws on a qualitative dataset that involved interviews with health system stakeholders involved in the response to the Covid-19 pandemic in Colombia,

## Abstract

This paper examines the role of temporality in the negotiation of unplanned adaptive tasks that were part of the health system response to the Covid-19 pandemic in Colombia. While research has been carried out on the temporal aspects of emergency preparedness, we argue that there is an empirical gap concerning how health care organizations responded with temporal urgency to Covid-19. The dataset (118 interviews) from which a subset of interviews were analysed for this paper was collected during the first wave of the pandemic in Colombia in 2020. Interviewees included representatives of national and regional governments, public hospitals and private clinic managers, clinicians, including physicians and nurses, laboratory directors, and academics. Narratives of two tasks are presented: reconfiguring clinical laboratories to expand the testing capacity for Covid-19 and increasing intensive care unit capacity for patients hospitalized with Covid-19. Through thematic analysis of the navigation of these tasks, the concept of "temporal shifts", which signifies how organizations use time as a resource (analogous experiences, future projections) to negotiate unplanned service changes, is developed. This study highlights how powerful stakeholders deploy past and future projections to influence others´ perceptions in the negotiation of temporal shifts: a type of change that differs from the incremental and planned types described in previous organizational literature on temporality. This shift was initiated by rapid task delegation via organizational hierarchy, but accomplished through pressured, improvised actions at the operational level. The policy and practice implications we suggest relate to addressing social and organizational effects, including consequences for stakeholder engagement and staff wellbeing, generated by organizational leaders making decisions under "time stress".

## Introduction

Learning from health system experiences related to Covid-19 has generated significant academic inquiry and managerial attention. In addition to the need to learn from perceived failings in health systems, some managerial research emphasizes a positive, unintended consequence of the pandemic: service change was accomplished at an accelerated pace in an

who represent potentially vulnerable staff at various system levels. The raw interview recordings and transcripts include potentially identifying or sensitive personal and organizational information. Following research ethics guidance relating to participant anonymity, we are not able to disclose the raw data. However, the minimal dataset, with regard to types of interview participant and relevant quotations derived from the interviews, are included in anonymised form in the manuscript. All relevant data are within the manuscript and its Supporting Information files.

**Funding:** The study from which this paper is derived was funded by the Colombian Ministry of Science, Technology and Innovation (Minciencias), grant number 1204101577001 (recipient Dr. Simon Turner). The funders had no role in study design, data collection and analysis, decision to publish, or preparation of the manuscript.

**Competing interests:** The authors have declared that no competing interests exist.

urgent context, as typically competing stakeholder interests were aligned [1]. This paper engages with social constructivist perspectives on the role of temporality in the navigation of organizational processes for understanding change processes in emergency contexts such as pandemics. Temporality is understood as the construction and communication of subjective time to influence actions, from organizational and policy leaders announcing a crisis through to setting the pace of responsive activities [2].

While research has been conducted on the temporal aspects of emergency preparedness, we argue that there is an empirical gap concerning how health care organizations responded with temporal urgency to Covid-19. From an institutional perspective, health systems have a variety of stakeholder interests that maintain existing "temporal structures" that are likely to constrain accelerated service changes [3]. From an emergency preparedness perspective, the pandemic was unanticipated by health systems internationally, suggesting that time pressures were navigated via improvised responses, rather than by putting into practice pre-established plans [4]. This paper argues that accelerated responses by health systems to the pandemic sat outside these perspectives. Temporal structures deviated from their "everyday" rhythms despite being situated in the institutionally conservative context of health services, while the unanticipated nature of the crisis precluded the navigation of time pressures using an "engineered" or pre-planned capability.

This paper asks the following question: in what ways are temporal resources employed by health system stakeholders to negotiate unanticipated adaptive tasks? First, we posit that stakeholders´ perceptions of both future and past states appear relevant in urgent contexts: a sense of an impending emergency was a motivating force for action, while reflecting on past experiences was a resource for navigating present actions. Second, we propose that these two forms of temporality, perceptions of past and future, are mobilized by stakeholders to influence the approach to, and trajectory of, change (e.g. managers could communicate urgency to encourage the reaching of consensus on time critical decisions). Finally, we argue that the deployment of temporal resources maintained, rather than challenged, existing institutional interests in Colombia, as the critical Covid-related tasks we studied were assigned and managed via existing organizational hierarchies.

By utilizing stakeholder narratives of two critical tasks undertaken within the Colombian health system to respond to Covid-19 –expanding laboratory testing capacity and scaling up appropriate Intensive Care Unit (ICU) space–this paper analyses the relevance of temporality (both past and future projections) for navigating these tasks and how stakeholders mobilized ideas associated with this concept to steer and influence change processes.

## Temporality and health system responses to Covid-19

In the health services research and policy literature on Covid-19, temporal concepts are relevant to the analysis of health system reactions in several ways. First, innovation processes to mitigate the effects of the pandemic were accelerated [1]. In the Netherlands, a sense of urgency and shared purpose promoted non-competitive behaviour among a variety of interacting organizations, accelerating the collaborative development of adapted ventilators for treating Covid-19 patients [5]. Second, time became a malleable concept in the communication of the development of innovations. The reporting of vaccine development in the UK and the US delivered a narrative of assurance to overcome concerns about "rushed" products, by pointing to a lineage of scientific practice relating to vaccine development that predated Covid-19 [6]. Third, temporal concepts were invoked in governmental political discourse in managing responses to the pandemic. The Finnish government´s publicly communicated stance on policy responses was one of "epistemic humility"–or retaining political flexibility in

the face of the unknowability of the future–in acknowledging the uncertainty associated with the virus´ trajectory [7]. In the UK, the Johnson government drew on temporal constructions when communicating with the public, making links to past crises, while remaining open to multiple imagined states, to justify and obtain support for the government´s evolving–or more pointedly indecisive–response to the pandemic. [8]

## Temporality from a management and organization perspective

Social constructivist perspectives on temporality as applied to management and organizations offer a conceptual language that can help to make sense of the analysis of health system responses to Covid-19. Among constructivists, there is debate about the relevance of institutional change relative to managerial processes in influencing the temporal rhythms by which critical tasks are accomplished. One perspective emphasizes the institutional context, arguing that temporal processes exhibit relative stability over time, as they become embedded in taken-for-granted social practices and norms. The "everyday" nature of temporality–including time-orienting artefacts such as calendars, scheduled meetings, and common periodic deadlines–become the mundane background of workplace organizing [3]. Within this perspective, the challenge for professions and organizations lies in changing the predominant framing of temporal processes (e.g. attitudes towards setting and meeting deadlines), which is said to involve making incremental changes to "everyday" perceptions of temporality. However, this is constrained potentially by multiple stakeholders who need to collectively shift their temporal framing of organizational processes. This "everyday" perspective–where change is conceptualized as a slow and incremental process–faces difficulties in explaining how organizations can make rapid adaptations in the context of temporal uncertainty [4].

An alternative constructivist perspective on responding to temporal change emphasizes the relevance of managerial processes undertaken by organizations to mitigate temporal uncertainty before "the event", e.g. investment in capabilities that support planned resilience [9]. An ethnographic study of firefighting operations in Hamburg illustrates how previous investments by the firms (including training to implement initial triage routines and using "preestablished rhythms" to control the timing of collective actions) provide "temporal autonomy" or "getting ahead of time" when responding to emergency incidents [4]. Moreover, "fast organizations", such as medical trauma centres, are said to rely on previous investments in coordination with the aim of producing time critical, and error free, responses when addressing events characterized by temporal uncertainty [10].

The environmental shock represented by the pandemic does not fit neatly with either of these constructivist perspectives on temporality ("everyday" or "planned"). From the "everyday" perspective, the process of incremental change appears to be inadequate for responding to an immediate environmental shock. Temporality is treated predominantly as a mundane organizing device, one that typically undergoes only incremental change through shifting human practices and orientations. The relevance of external, environmental shocks–as exemplified by the 2020 pandemic–in shifting concepts of time, in potentially abrupt and radical ways, has been neglected. Empirical evidence is needed on the ways in which temporal concepts, e.g. heightened urgency or impetus for a reaction, affect change processes in the context of environmental shocks.

The second constructivist perspective, which emphasizes managerial planning and adaptation, assumes that organizations make previous investments in capabilities–so-called planned or precursor resilience [9]–to address contexts of temporal uncertainty (e.g. the activities of emergency services). However, the relevance of this perspective to planning and provider organizations within health systems facing the Covid-19 pandemic–recognized internationally as a

novel problem for which health systems lacked preparedness–is questionable. Health service organizations are traditionally regarded as conservative settings–composed of uni-professional groups that tend be resistant to change [11]–with the metaphor of "fast-moving" organizations [10] applying more narrowly to known categories of emergency care, as health system change tends to be informed by earlier rounds of planning and organization. In England, for example, hyper-acute stroke units that are dedicated to, and set up for, rapidly diagnosing and treating suspected stroke patients were the product of evidence-based planning and stakeholder consultation processes in specific metropolitan areas undertaken over a period of years [12]. Health system responses to Covid-19 took place against a different temporal background to emergency care. As the pandemic was largely unanticipated and not planned for, the arrival of this external, environmental shock suggests the need for health system organizations to undertake a "temporal shift", or change in pace or "tempo" [2], to respond to the immediate and emergent pressures presented by Covid-19.

To undertake a "temporal shift", upon which resources might health service organizations draw? If precursor or planned resilience is limited in responding to an unanticipated event, then organizations need to construct resources during the event–or "in the middle of things" [13]–using features of the social and material context to support problem-solving. Such improvised responses have been termed "bricolage" [14]. However, improvised responses are likely to face institutional constraints in conservative contexts such as health care, including a social structure of entrenched and differentiated interests, that requires negotiation to allow the introduction of newly agreed upon practices. Health system responses to Covid-19 were improvised, collective changes taken in reaction to a novel problem that outpaced "everyday" (mundane) temporal rhythms, but were not the product of "engineered" (planned) strategies to address the temporal uncertainty of known emergency contexts. This paper explores the temporal resources used by health system stakeholders to navigate, and shape the coordination of, critical tasks associated with Covid-19, including the employment of past and future projections as organizing devices.

Other theoretical perspectives, notably historical institutionalism, can be applied to the study of temporal change. For example, historical institutionalism recognizes tensions between past arrangements, due to path dependency, and path-bending forces associated with external environmental shocks or "critical junctures" [15]. For historical institutionalists, the pandemic represented a window of "political openness" for accelerated governmental decision-making on policy change [16], although the consensus appears to be that radical change was moderated by maintenance of the existing institutional context, including structural interests [17], continuity in the dominant ideological or policy paradigm [18], and flexible use of existing systems [19]. The evidence suggests the gradual evolution of existing policy ideas to meet identified public policy challenges of the pandemic rather than a radical disjuncture associated with the Covid-19 crisis [20–22].

These factors are relevant to the social constructivist perspective adopted in this paper, particularly in highlighting the importance of the macro public policy context in shaping responses to crises such as the pandemic. Using historical institutionalism to analyse macro level policy actions is beyond the scope of the current study, although we advocate in the discussion for its combination with social constructivism in future research.

## Materials and methods

### Research context

This paper is derived from a wider qualitative study of health system responses to Covid-19 in Colombia [23]. A qualitative design was chosen because it is well suited for capturing

stakeholders´ perceptions and experiences in relation to a novel context. Colombia, an upper-middle-income country in Latin America, has a social insurance model of care with contributory and subsidized regimes, that includes publicly and privately run providers, for accessing general and specialist health services [24]. Colombia was severely affected by Covid-19, experiencing the fourth highest number of deaths in the region following Brazil, Mexico and Peru [25]. Colombia was chosen as a case study because it navigated unplanned managerial and policy actions in response to Covid-19 [26], raising questions about how it undertook the temporal shift necessary to respond rapidly in an emergency context, despite suffering historically from a lack of human capital resources, limited infrastructure, budgetary constraints, including high debts between actors, and system fragmentation [27, 28].

Key organizational factors in Colombia´s response to the pandemic identified in previous literature include shared purpose that propelled formal [29] and informal cooperation within the health system [30], inter-sectoral collaboration at the local system level [31, 32], and the repurposing of organizational and professional roles to redirect resources toward meeting emergent strategic goals [33]. This paper builds on these findings by using micro level concepts (the enactment of temporal urgency and historical learning) to analyse how professional and organizational interactions were motivated and negotiated in service of the rapid health system response to the pandemic across two service areas in Colombia. This approach complements previous studies by describing interactional mechanisms through which rapid changes were achieved, while also tracing unintended consequences of the accelerated approach to change, using micro level concepts that capture how and why the temporal context of the pandemic influenced change processes.

## Data collection

The dataset (118 interviews) from which a subset of data was analysed for this paper was collected during the first wave of the pandemic in Colombia in 2020. The interviews were conducted from 11 June 2020 to 13 November 2020. Participants were purposively sampled to provide a variety of stakeholder perspectives on health system responses to Covid-19 by Colombian national entities and in the major cities of Bogotá, Cali, and Cartagena. We chose two cases, the scaling up of laboratory testing and transformation of ICU capacity, as these tasks (expanding testing to control spread of Covid-19 and providing intensive care for severely affected patients) represented national and international priorities in response to Covid-19. Data from 15 interviews used in this paper (S1 Table) related to participants able to provide an insider´s perspective on the scaling up of laboratory testing or ICU capacity in response to Covid-19, including representatives of national and regional governments, public hospitals and private clinic managers, clinicians, including physicians and nurses, laboratory directors, and academics.

The study was carried out in accordance with the consolidated criteria for reporting qualitative studies [34]. Participants were invited by email, received an information sheet, and provided informed written consent for participating in interviews. The interviews were completed by employed researchers with postgraduate social science training. Participants were informed about the purpose of the study and how their involvement in an interview about their perceptions and experiences of responses to Covid-19 supported the study´s objectives. All interviews were conducted virtually in Spanish and digitally recorded using web-paged meeting platforms, including Microsoft Teams and Zoom. The interviews lasted approximately 1 hour; the discussion was semi-structured using a topic guide (S1 Appendix), audio-recorded and transcribed professionally. Prior to the commencement of data collection, ethical approval was provided by the Committee on Research Ethics, Universidad de los Andes; it was classified as low risk.

## Data analysis

Given our research interest in stakeholders'use of temporal resources to navigate non-routine tasks, we were attentive to how past experiences and imagined futures were perceived by stakeholders and influenced approaches to pandemic-related activities (laboratory testing and ICU capacity). The data were analysed in two stages drawing on deductive and inductive approaches [35], whereby manual coding of the data by the two researchers was informed both by organizational literature on temporality (deduced from existing concepts, e.g. past and future projections, temporal structuring and uncertainty) and by attending to the unique perceptions and experiences of the participants (inductive). The process of thematic analysis was as follows. First, narratives of the processes of adapting two aspects of Colombia´s health system in response to the pandemic were constructed: (1) expanding laboratory testing capacity and (2) creating dedicated ICU spaces for patients with Covid-19. For each process, we drew on interview data to describe the problem facing the health system, the key actions taken to address this problem, including those that came due to regulations drawing on relevant policy documents, and the provisional outcomes from these actions. Second, the processes involved in navigating the tasks were extracted from the interview data using two themes: enacting temporal urgency (sub-categories; how the severity of the situation was mobilized, influencing motivation to participate, and pragmatic actions taken) and enacting historical learning (sub-categories; how past experiences, relationships, and perceived system failings were mobilized to gain influence in the current response). Analytic data saturation was reached when a full account of the temporal process of navigating the two tasks was reached. A coding tree of the themes by task is provided in S2 Table.

## Results

### Adapting and scaling up laboratory testing capacity

**Enacting temporal urgency.** The case narrative is presented in S3 Table [36, 37]. The laboratory case was characterized by urgency of change: rapid adaptation was demanded and celebrated by senior-level planners and providers of laboratory services, yet responsibility for enacting such calls for urgent action appeared to be delegated to operational-level staff with piecemeal organizational support. While strategic decisions were often taken quickly and at a high level, the pace of decision-making appeared to preclude analysis of implementation practicalities. At the governance level, it was felt that there was a lack of communication and support from university management in converting the laboratories to enable the testing of samples for Covid-19. In this operational manager´s experience, the senior management of his university instructed him to "go ahead" with setting up the laboratory for testing, but he felt that the urgency with which it was suggested that he proceed was not accompanied by the necessary organizational support:

> "They simply told us, "well, the agreement has been signed". So, "yes, go ahead", but we didn´t even know what the agreement meant. I think that the university failed to create a committee that met constantly, not only of us who were like those at the bottom, those who worked, but to pass on that information to the directives. And I think that was a great failure of the university" (SH-B-30, university academic, Bogotá).

Despite the perceived lack of support, the pressing, and life-threatening nature, of the pandemic motivated action at the operational level. Laboratory managers perceived a range of motivations, from altruistic through to competitive incentives. As an example of altruism,

this public laboratory manager explained that no charges would be made for testing samples by their laboratory, as the additional infrastructure represented a "gift" to society:

> "there is one thing I made clear to us at the beginning, we don´t charge anything, I offered God that gift, not one peso for processing the samples from the Departmental Laboratory, not one peso. We do provide supplies, we do provide personnel, we do provide premises. . . but we give away that work. They bring RNA, they bring their coronavirus kit; but we put on the gloves, we put in the other things, all the supplies, us. So that is a gift from [the laboratory] to society" (SH-D-025, Laboratory representative, Cartagena).

In contrast, another interviewee described competition among the converted laboratories to gain recognition for being the first to be accredited to test samples, noting how jealousy among institutions undermined collaboration in this pursuit:

> "we perceived it as if everyone wanted to compete and not collaborate, and it was a competition to see who did it first and who made a fuss in the media. During that time, it was like that—the one who said it first was the one who was taking the credit. And so, it started that game of jealousy between the different institutions. It seemed to me that there was a lot of jealousy" (SH-D-24, University director, Cartagena).

Whether motivated by altruism by contributing to society, or driven by self-interest in being seen publicly to contribute to a high-profile national emergency, the temporal threat to life presented by the pandemic prompted urgent actions to adapt and scale up laboratories for Covid-19 testing at the operational level.

**Enacting historical learning.** The past was enacted in particular ways by stakeholders in describing responses to Covid-19. For some, the past encompassed a set of experiences that was used as a resource to respond to, and gain influence over, the current situation. The interview participants emphasized the relevance of the previous experience and particular skillset that their profession or organization were able to bring to the problem. The following interviewee emphasized the laboratory´s 15-year experience in conducting molecular testing. This capability allowed the organization to act quickly–or be "ready to go"–when the call to expand the national network for laboratory testing was made:

> "we had the great advantage that molecular methods were not something alien to our daily work. We have been doing molecular tests for about 15 years, mainly in transplants and more recently in molecular genetics, sequencing, molecular pathology and molecular infobiology. Then, when the news came that it had to be implemented and they were calling on private laboratories to shake hands with the laboratories of the national network of reference laboratories, such as the national institute for health and the district and departmental ones, we were ready to go" (SH-C-025, Laboratory representative, Cali).

In addition to technical skills, the past had enabled the accumulation of relevant relational capital that could be exploited in the current crisis. The agility of laboratories was also supported using latent relationships, both formal and informal. The importance of a formal collaborative research link–forged with an international research body in response to previous Zika outbreaks in Colombia, including that of 2015 –had been an ongoing source of research funding and knowledge, which had become a stronger relationship with time:

"they began to see that we were not third world researchers over there, we were not. You know, there is a phenomenon that is very complex of countries being data collectors and, personally, that bothers me horribly. When they first arrived, I clearly told them "we are international-level researchers here". In other words, we may look like Colombians here, but we are tough. And they were very respectful and I love [this research agency] because unlike other "gringo" agencies, [that research agency] is a respectful agency" (SH-A-018, National health agency representative, Bogotá).

Reflecting the concept of collaboration extending beyond formal institutional relationships described earlier, the value of this formal research link was couched in personal terms, as the relationship with the agency rested on mutual respect, allowing the two partners to become "friends", a factor that had contributed to the durability of collaboration. Another interviewee explained how the use of their personal friendship network supported day-to-day problem-solving in the laboratory, suggesting that this had been more important than institutional support which had, and continued to, present challenges:

"what is a bit tremendous is that this is personal, isn't it? It's because they are my friends; it's not institutional. No, this friend A, calls friend B: "hey, I'm scrubbing in the lab, help me"! Well of course, yes, obvious. But if they communicate on an institutional level, that never happens" (SH-C-24, laboratory director, Cali).

The past was also narrated as a locus of failings, signifying a need for new approaches to change during the pandemic. At the national level of the health system, collaboration among entities was said to be weak historically. This state was linked both to distrusting relations between different institutions and to the primacy of competitive individualism. One interviewee stated that, due to a lack of collaboration, "all the laboratories had to learn alone", linking this situation to a "Latin gene" which promoted individualism and a lack of openness towards others:

"the exchange of knowledge would have been better. . ., but this Latin gene that we have is terrible for what knowledge exchange is, this Latin gene is very closed, it doesn't allow one to talk to other peers" (SH-D-024, University director, Cartagena).

This tendency for individualism could be said to reflect "formal" relationships within the health system, that is, with the "other peers" not linked through personal networks who might be seen as "outsiders". More "informal" and collegial relationships among networks of individuals across entities–often based on personal ties including friendship–appeared to compensate for, and may have emerged in response to, this institutional distancing or "standoffishness".

While these institutional conventions set the context for interaction within and among entities making up the health system, different behavioural responses to this context were described. First, resources tended to be shared among informal networks within and across organizations, including friendship ties. Second, some interviewees emphasized alternative logics of production, centred more on openness and altruism towards others, as a reaction seemingly to the dominant institutional logic of individualism. Responses to the pandemic were influenced by, and to some extent challenged, this initial state. While the urgency of the situation could have precluded collaboration, the severity of the situation was a spur for new forms of collaboration, based on a mutual lack of experience to face this novel problem, and fear and anxiety of acting alone in response to the threat to life posed by Covid-19.

At the local service delivery level, experiences of collaboration prior to the pandemic were narrated in particular ways by interview participants. Perceived failings in past ways of

working were used to justify new approaches. A representative from a national research entity emphasized the need for "flexibility" in responding to Covid-19, an approach that contrasted with the entity´s "old" ways of working (SH-A-018, National health agency representative). Another interviewee highlighted challenges they perceived in past ways of working in the city where they were working, suggesting that the hierarchical style and slower pace of work was a "culture shock" relative to their former place of work:

> "I had a culture shock here when I arrived from Medellin here to Cartagena, because here you work with another rhythm, with other dynamics [. . .] "They were always saying, "Oh, doctor, but you see, you know, we have to authorize and the authorization then implies building a document, we have to have this, we have to have that, I don't have the time in the work plan", or rather, thousands of things [. . .] there is no time" (SH-D-24, university director, Cartagena).

S4 Table summarises consequences of the accelerated approach to scaling up testing capacity.

## Evaluating the role of temporality in adapting and scaling up ICU capacity

**Enacting temporal urgency to motivate and guide actions.**   The case narrative is presented in S5 Table [38–41]. A legal decree was introduced by the national government to support the scaling up of ICU capacity. The decree set out a national target, and new legal freedoms, for increasing ICU capacity, including the allocation of responsibilities to different institutional actors across the system to support this process. The instantaneous nature of this legal declaration, alongside the seemingly continual emergence of other protocols delivered top-down by a range of national level entities, meant that reviewing regulations–including working through perceived "contradictions"–became a key part of providers´ daily activities in responding to Covid-19:

> "The same happens with issues that appear within the same government, and sometimes contradictions also occur because the speed of the protocol made by the National Institute of Health, the Ministry of Health itself, or the [local] departmental secretaries at a given moment can be contradictory or one becomes outdated because it does not keep pace with the others" (SH-A-007, national research coordinator, Bogotá).

Some participants referred to future projections–some recounted from previous weeks and months, others constructed during interview–about the catastrophic consequences of reaching the limits of ICU capacity. The projected threat to life of this future state was described as a "fear" and "panic" inducing spur for collaboration:

> "There was even speculation that the managers had ICUs in their homes because the ICUs were not going to be enough, because there was only the capacity for so much, and that when that got saturated, the dead would be left on the streets. . . and well. . . all that film that wasn't a lie, but it wasn't so true either, but there was a fear and a panic about that situation, which forced us to say: "let's work together. Because here, if we don't act together, we are dead". So that's what I think happened internally to each of us, and each of us has gone out and given." (SH-D-11, Laboratory representative, Cartagena).

**Enacting historical learning.**   While a pandemic on the scale of Covid-19 was not anticipated directly, related preparations for emergencies were enacted, including plans for responding to another type of disaster (an earthquake) by a local health planning entity:

"I think it has been a good experience to test all the response systems we had and it is something completely new, for a long time I was also preparing, for example, the city for an earthquake [. . .] and a completely unexpected emergency came to us. We did not expect this to happen" (SH-B-018, Representative, commissioning organization, Bogotá).

However, a lack of relevant experience for undertaking the adaptation and expansion of ICUs came across strongly in our interviews with hospital leaders. The actions that were chosen (e.g. reorganizing hospital staffing) were characterized by uncertainty and undertaken with trepidation:

"Clearly, we had to go through many crises. Well, the first ones who began to go through the crisis were the ones who had to organize the institution because, first, we didn't know how, there was nowhere to take experiences, but suddenly what had happened in Europe and some of what we were already hearing from the United States, but we had nothing to take from experience and logically that caused us a lot, a lot of anguish, because you don't know if what you are preparing is correct or not." (SH-D-028, Hospital physician, Cartagena).

While the types of analogous experience recalled above were deemed relevant in responding to the pandemic, there was also recognition of the insufficiency of planning or experience accumulated before the event, as these were of little relevance when immersed in the event and needing to take situated actions "in the moment", as this interviewee explained when trying to match the potential capacity needed to meet an unknown, but projected increase in, demand:

"The issue is to apply at this time in high volume, if there is one, what we are handling as a preventive issue and we are ready with a lot of staff ready and with a lot of infrastructure ready and. . . well. . . you can prepare a lot for combat, but until you're faced there in the trench of things, you don't know how we're going to do it." (ES-B-003, hospital director, Bogotá).

Similar to the laboratory testing case, the adaptation and scaling up of ICUs drew on latent relationships, especially connections that were pipelines for support across different geographical localities. In the example below, some staff were seconded to a clinic in a different city with an insurer´s network to the support the absorption of new clinical protocols:

"[The insurance company] has a clinic at the university, in which they have been working on training courses, but despite that, that is one of the things that has been done. We have moved personnel, an important point that I think is a good experience, for example as the pandemic and the speed of the pandemic has been different in the different cities. For example, he made our personnel go to Barranquilla to mix qualified personnel with new personnel to learn and manage the protocols" (SH-A-007, national research coordinator, Bogotá).

An educational link–in the form of a request for support by a former student–was also used to provide support for responding to Covid-19 by an indigenous community located in the Amazon:

"I began to remember that we have a graduate who is from an indigenous community in the Amazon [. . .] I called her and she said "here we don't know anything about COVID."

So, I sent her information about COVID. And, later, she sent a video in which she translates what she has read into the language of her indigenous community and she begins to explain to them what COVID is. And that was very nice, and she is the one who gives me all the data to start making the connections with the Amazon." (SH-A-001, Nursing association representative, Bogotá).

S6 Table summarises consequences of the accelerated approach to scaling up ICU capacity.

## Discussion

A summary of the results is provided in S7 Table. Decision-making on introducing change was navigated rapidly and led top-down, while responsibility for delivering change was delegated to operational staff. There were concerns among some interviewees that the pace of top-down decision-making on introducing change limited stakeholder involvement (e.g. nursing staff representation in ICU care) and left various implementation considerations (e.g. human resource capacity, training, and wellbeing) unresolved. Improvised actions by staff at the operational level (e.g. personal network use) addressed gaps in implementation generated by the top-down approach to change. Staff responsible for task delivery used the past as a resource for negotiating the tasks, including knowledge derived from relevant experiences and the rekindling of latent relationships for sharing information and physical resources.

### Research implications

Covid-19 represented an alternative category of problem, and consequent set of challenges for health system planning, compared with those represented in much of the previous literature on temporality. It required rapid responses that needed to outpace "everyday" temporal structures [3], while being widely under anticipated by health systems, which largely precluded emergency preparedness [4]. The communication by leaders of severe urgency facilitated a temporal shift in the tasks analysed by circumventing institutional constraints typically seen in multi-stakeholder contexts such as health care [3]. This study contributes to the temporality literature by showing how the communication of urgency in extreme situations acts as a temporal resource for coordinating change that overcomes predicted institutional constraints, serving to undo the basis for concerns that might otherwise come from "lower level" or operational actors.

Interactions between power and context in the mobilization of temporal resources should be recognized. Our study revealed how hierarchical authority combined with temporal resources to wield power, seen as the ability to influence others´ behaviour in organizational contexts [42]. Past and future projections shaped the approaches to the critical tasks we analysed. In relation to the future, leaders enacted a context of urgency to stimulate rapid and intensive actions by those under their authority. In the laboratories case, university executives, and the national entity overseeing the testing programme´s expansion, were able to place pressure on laboratory staff directors to pursue accreditation processes, including weekend national travel for training at the pandemic´s onset, by signalling the urgency of the need to fix the testing infrastructure to avoid impending crisis.

Some interviewees´ accounts of this process suggest that not much of an argument or case was made to communicate this urgency. Directives were issued remotely (e.g. by email) and few words were used to delegate responsibility and engender ownership among the recipients of hastily shared orders. In relation to the past, the narration of perceived failings in the existing health system was used to legitimize alternative approaches to the coordination of critical tasks (e.g. shifting responsibility for emergency ICU funding from national level insurers to

local care commissioners, as the former entities were accused of failings historically in reimbursing hospital costs).

Leaders´ use of temporal resources in extreme contexts represents a distinctive approach to health system change. Previous research presents health system change as a social and negotiated process, whereby those leading change rely on evaluative evidence [43] or "political skill" [44] to convince others of an innovation´s worth. While the influence of political or evidential resources is entangled with power structures (e.g. powerful clinicians´ use of complex publications to steer decision-making on innovation) [45], change has been depicted as a negotiated process in which a variety of stakeholders can exert influence (e.g. hospital managers´ counter use of local audit data in the study just described). Health system change under temporal pressure appears to shortcut this negotiated political process. The approaches to the "innovations" we studied lacked evaluative evidence [43]; their implementation was associated instead with leaders´ use of (a) time pressure as a proxy for evidence and (b) the deployment of hierarchical authority to hastily delegate actions to implement the desired outcomes.

In contexts of extreme time pressure, leaders´ use of temporal resources appears to motivate change in place of an evaluative–and negotiated–evidence base that is typically needed to mobilize change in health care systems as multi-stakeholder contexts. This finding speaks to concerns among social constructivists about using concepts of power to explain the temporal structuring of social interactions [3]. In the critical tasks studied in this paper, leaders´ ability to mobilize temporal resources relied on hierarchical power structures associated with traditional authority and accountability relationships. For example, the expansion of ICU capacity was coordinated top-down via a legal decree that set out a national target for expansion, changes to financing, and prescribed organizational roles. The consequent social interactions around implementing the decree were then characterized by (a) the assertion of urgency by mid-level organizational leaders based on the top-down call for accelerated change, (b) temporally pressured forms of working and organizing, and (c) rounds of reinterpretation work at the operational level to align policy dictates with clinical realities.

Social constructivists need to stitch managerial and institutional power bases into their analysis of social interactions to explain the maintenance of temporal structuring as well as temporal restructuring of task coordination. One promising method is to combine insights from macro historical institutionalism with micro level social constructivism. For example, deployment of temporal resources at the micro and meso levels may play a role in shaping policy responses at "critical junctures", such as during the Covid-19 pandemic. In this study, temporal resources appeared to have a moderating effect on new approaches to change. Decision-making on change was accelerated, but not more egalitarian at the juncture we studied. This indicates that leaders´ temporal resource use (e.g. communication of urgency in task delegation) maintained existing institutional interests. The use of hierarchical authority allowed leaders to engage in rapid task delegation and to raise the tempo of work across hierarchical tiers. Organizational hierarchy, and the exercise of positional power this allows in coordinating task and authority relationships, is an additional moderating mechanism for explaining continuity in institutional interests at critical junctures such as pandemics.

The directive approach to change identified in this extreme context prompts us to question the celebratory tone that tends to feature in analyses of fast-moving organizations´ navigation of temporal uncertainty [10]. Temporal shifts, as exemplified by the Covid-19 pandemic, are not experienced in the same way universally but are grounded in context specificities. The relevance of managerial guidance based on high-income economies´ experiences of Covid-19 (e.g. the "system shock framework" tested in Australia [46]) is blunted by variation in contextual antecedents found in different health systems. Resource pressures, including system fragmentation, facing Colombia and other health systems in Latin America, were noted at an early

stage of the pandemic [47]. Our study showed that the temporal shift needed to introduce rapid changes in the functioning of Colombia´s health system was accompanied by concerns about staff wellbeing, uncertainty over financial reimbursement, ad-hoc information sharing, and significant personal goodwill by individuals to absorb additional roles and responsibilities. Temporal shifts interact with other contextual variables in processes of change beyond institutional stakeholders, including funding sources, governance conventions, resource pressures, and workforce capacity. International evaluations of unplanned temporal shifts need to consider a range of interacting contextual variables within idiosyncratic health systems.

## Policy and practice implications

The implications of this exploratory, qualitative study for managing temporality should be treated with caution. Our limited cases suggest that the individual psychological "time-stress" [48] of navigating decision-making under time pressure has wider social and organizational consequences. The effects in the cases presented here included: the rapid approach to decision-making that constrained stakeholder involvement; leaders´ communication of the urgent threat to life, which hastened implementation by treating concerns about change at lower levels as "resistance" to be overcome; and the intensity of work needed during implementation, which affected the wellbeing of staff at operational levels. The policy and practice implications we suggest relate to addressing these social and organizational effects generated by organizational leaders making decisions under "time stress".

First, despite the need for swift action in response to emergency contexts, it is important to involve stakeholders–at an early stage—in decision-making on service change, to aid in the anticipation of challenges in implementation as plans are rolled out. Managers should ask: who is in the room who will be affected by this decision, and who is currently missing due to perceived time constraints? Patient and public involvement mechanisms need to be carefully planned and structured in advance–taking into account potential imbalances in power, diversity, and social inequities [49]–to regulate the mix of actors informing "co-production" in times of temporal stress. Temporal concepts also point to the value of bringing a variety of past and current experiences to bear on complex problems characterized by uncertainty and time pressure. Stakeholder engagement mechanisms–tailored to pandemic use–should be planned in advance.

Second, health system experiences of Covid-19 suggest that emergency preparedness–an aspect of planned or precursor resilience–could be extended to cover a wider scope of activities, making a dent into tasks that are otherwise left for negotiation–in the moment of emergency–through situated actions that rely on improvisation or "bricolage" [14]. In this study, the past was mobilized as an informal temporal resource in actors´ responses, from using experiences of analogous emergencies through to activating personal networks. The implication for management is about evaluating which sources of temporal uncertainty can be anticipated, and prepared for, such that they become a feature of "everyday" temporal structures. Managers should ask: what more could we have anticipated, earlier, in our routine practice to save future time consumed by avoidable improvisation? Barriers and enablers to durable organizational learning from pandemics–which involves the translation of experience into standardized routines for guiding action and ongoing feedback and evaluation [50]–should be analysed in specific organizational settings by practitioners and researchers alike. Health system leaders should ensure experiential learning from the pandemic is codified to inform future emergency planning.

Third, managers should reflect on the value of the narratives they deploy relative to other forms of leadership needed to motivate others in emergency contexts. In the task-specific

responses to Covid-19 analysed, a variety of motivating impulses among the actors involved in the response were revealed, ranging from fear and anxiety about avoiding a catastrophic future through to actions hailed as "heroism" for absorbing additional roles. Preoccupation with motivation reflected the intensity of the tasks faced that breached knowledge, capacity, and emotional limits. The availability of managerial support for navigating delegated tasks was more piecemeal; as recommended for low- and middle-income countries more broadly [51], long-standing investments in workforce training, financing, and motivation (both words and deeds) is needed to face future health emergencies in Colombia. Managers should ask: how do my words and deeds affect the motivation and wellbeing of my staff when collectively confronting tasks with temporal uncertainty? More "open" and empathetic forms of leadership [1]–based on acknowledging the human value of, and navigating tasks in close coordination with, mid- and operational- level staff–represent one way of countering negative effects upon staff in periods of temporal stress. Health providers´ experiences during Covid-19 should inform emergency planning for future pandemics to mitigate the knowledge, capacity, and emotional burden associated with avoidable ad hoc negotiation of tasks during crises.

## Conclusions

This study highlights how powerful stakeholders deploy past and future projections to influence others´ perceptions in the negotiation of "temporal shifts", a type of change that differs from the incremental and planned types described in previous organizational literature on temporality. This study has limitations that point to the need for further research. The findings were derived from one context, a health system from the Global South, which limits the generalisability of these findings to other settings. Future studies should compare the role of temporal concepts, such as "urgency" and "history", in response to emergencies like pandemics across different geographical and economic sectors, e.g. comparisons of health care with education [52]. The data represent initial responses to the pandemic in 2020; longitudinal studies could examine what influences the institutional "refreezing" [53] of health systems in new states (e.g. approaches to collaboration) or the reasons why aspects of care planning and delivery revert to pre-crisis states.

## Supporting information

**S1 Appendix. Interview topic guide.**
(PDF)

**S1 Table. Interview participants.**
(PDF)

**S2 Table. Coding tree.**
(PDF)

**S3 Table. Laboratory case narrative.**
(PDF)

**S4 Table. Consequences of accelerated change (laboratories).**
(PDF)

**S5 Table. ICU case narrative.**
(PDF)

**S6 Table. Consequences of accelerated change (ICUs).**
(PDF)

**S7 Table. Summary of findings.**
(PDF)

## Author Contributions

**Conceptualization:** Simon Turner.

**Data curation:** Simon Turner, Dennys Paola Fernandez.

**Formal analysis:** Simon Turner, Dennys Paola Fernandez.

**Funding acquisition:** Simon Turner.

**Investigation:** Simon Turner, Dennys Paola Fernandez.

**Methodology:** Simon Turner, Dennys Paola Fernandez.

**Project administration:** Simon Turner.

**Writing – original draft:** Simon Turner, Dennys Paola Fernandez.

**Writing – review & editing:** Simon Turner, Dennys Paola Fernandez.

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
