## [Decision Letter · Decision Letter 0]

2 Jul 2024

PONE-D-23-30367“Everything was much more dynamic”: temporality of health system responses to Covid-19 in ColombiaPLOS ONE

Dear Dr. Turner,

Thank you for submitting your manuscript to PLOS ONE. After careful consideration, we feel that it has merit but does not fully meet PLOS ONE’s publication criteria as it currently stands. Therefore, we invite you to submit a revised version of the manuscript that addresses the points raised during the review process.

Please note that we have only been able to secure a single reviewer to assess your manuscript. We are issuing a decision on your manuscript at this point to prevent further delays in the evaluation of your manuscript. Please be aware that the editor who handles your revised manuscript might find it necessary to invite additional reviewers to assess this work once the revised manuscript is submitted. However, we will aim to proceed on the basis of this single review if possible. 

The reviewer has recommended that you expand your literature review, and discuss your findings in relation to previous studies about the COVID-19 pandemic.

Could you please carefully revise the manuscript to address all comments raised?

We look forward to receiving your revised manuscript.

Kind regards,

Johanna Pruller, Ph.D.

Associate Editor

PLOS ONE

Journal Requirements:

5. We note that you have indicated that there are restrictions to data sharing for this study. For studies involving human research participant data or other sensitive data, we encourage authors to share de-identified or anonymized data. However, when data cannot be publicly shared for ethical reasons, we allow authors to make their data sets available upon request. For information on unacceptable data access restrictions, please see http://journals.plos.org/plosone/s/data-availability#loc-unacceptable-data-access-restrictions. 

Reviewers' comments:

Reviewer's Responses to Questions

**Comments to the Author**

1. Is the manuscript technically sound, and do the data support the conclusions?

Reviewer #1: Yes

2. Has the statistical analysis been performed appropriately and rigorously? 

Reviewer #1: N/A

3. Have the authors made all data underlying the findings in their manuscript fully available?

Reviewer #1: No

4. Is the manuscript presented in an intelligible fashion and written in standard English?

Reviewer #1: Yes

5. Review Comments to the Author

Reviewer #1: The paper discusses an interesting topic: the role of temporality in crisis responses, through a qualitative analysis of health system providers and stakeholders during the COVID-19 pandemic in Colombia. However, some changes are needed to improve its contributions. First, the paper adopts a specific theoretical discussion related to organizational studies, and some gaps are pointed out, but other approaches, such as historical institutionalism, discuss these gaps (the relevance of external environmental shocks and the importance of past experiences). Secondly, the paper should develop the findings of previous studies about the COVID-19 pandemic, including the COVID-19 pandemic in Colombia. Thirdly, the study briefly discusses the features of public policy (including the dynamics and legacies of Colombia's health care policy) and the role of politics and political leadership. It would be important to develop that as it would support the case contextualization and the variables that could influence the changes analyzed. Finally, it would be important to better grasp the contributions of the paper to understand other crises or the pandemic in other countries.

6. PLOS authors have the option to publish the peer review history of their article (what does this mean?). If published, this will include your full peer review and any attached files.

Reviewer #1: No

---

## [Author Response · Author response to Decision Letter 0]

10 Jul 2024

Dear Dr Pruller, many thanks for your decision letter and comments. Please see the point-by-point response to these in the file attached. Best wishes, Simon Turner

---

## [Editor Report · Decision Letter 1]

6 Sep 2024

PONE-D-23-30367R1“Everything was much more dynamic”: temporality of health system responses to Covid-19 in ColombiaPLOS ONE

Dear Dr. Turner, 

Thank you for submitting your manuscript to PLOS ONE. After careful consideration, we feel that it has merit but does not fully meet PLOS ONE’s publication criteria as it currently stands. Therefore, we invite you to submit a revised version of the manuscript that addresses the points raised during the review process.

We look forward to receiving your revised manuscript.

Kind regards,

Faten Amer, PhD in Health Sciences

Academic Editor

PLOS ONE

Journal Requirements:

Additional Editor Comments:

Dear Dr. Turner,

Thank you for your patience during the review process of your manuscript titled “Everything was much more dynamic”: Temporality of Health System Responses to Covid-19 in Colombia. Due to challenges in finding suitable reviewers, the review process took longer than anticipated. As a result, the editor has conducted a thorough evaluation of your manuscript.

Strengths:

The abstract and introduction clearly outline the study’s objectives and significance.

The comparative case study approach provides valuable insights into crisis management during the COVID-19 pandemic.

The discussion integrates findings with existing literature and highlights the importance of temporality in health system responses.

Areas for Improvement:

Length and Redundancy: The manuscript is notably lengthy, and some sections contain redundant content. We recommend revising these sections to make the text more concise. Please utilize the appendices more efficiently to manage detailed content.

Methods: Provide more clarity on case selection criteria and interview methodology. Expand on how data analysis was conducted, including coding and categorization of data.

Discussion: Break down long paragraphs and ensure clearer connections between findings and broader policy implications. Offer more detailed and actionable recommendations.

Clarity and Structure: Shorten long paragraphs in the discussion to enhance readability. Address overly dense text to improve reader engagement.

Please make the necessary revisions to address these points and resubmit your manuscript for further review.

Thank you for your attention to these revisions.

Best regards,

Dr. Faten Amer

---

## [Author Response · Author response to Decision Letter 1]

9 Sep 2024

Please see the cover letter for a point-by-point response to the review comments.

---

## [Editor Report · Decision Letter 2]

12 Sep 2024

“Everything was much more dynamic”: temporality of health system responses to Covid-19 in Colombia

PONE-D-23-30367R2

Dear Dr. Simon Turner, 

We’re pleased to inform you that your manuscript has been judged scientifically suitable for publication and will be formally accepted for publication once it meets all outstanding technical requirements.

Kind regards,

Faten Amer, PhD in Health Sciences

Academic Editor

PLOS ONE

---

## [Editor Report · Acceptance letter]

18 Sep 2024

PONE-D-23-30367R2 

PLOS ONE

Dear Dr. Turner, 

I'm pleased to inform you that your manuscript has been deemed suitable for publication in PLOS ONE. Congratulations! Your manuscript is now being handed over to our production team.

Kind regards, 

on behalf of

Dr. Faten Amer 

Academic Editor

PLOS ONE